# Dynamic breaking of a single gold bond

Ilya V. Pobelov[1,*], Kasper Primdal Lauritzen[2,*], Koji Yoshida[1], Anders Jensen[2], Gábor Mészáros[1,3], Karsten W. Jacobsen[4], Mikkel Strange[2,4], Thomas Wandlowski[1] & Gemma C. Solomon[2]

While one might assume that the force to break a chemical bond gives a measure of the bond strength, this intuition is misleading. If the force is loaded slowly, thermal fluctuations may break the bond before it is maximally stretched, and the breaking force will be less than the bond can sustain. Conversely, if the force is loaded rapidly it is more likely that the maximum breaking force is measured. Paradoxically, no clear differences in breaking force were observed in experiments on gold nanowires, despite being conducted under very different conditions. Here we explore the breaking behaviour of a single Au–Au bond and show that the breaking force is dependent on the loading rate. We probe the temperature and structural dependencies of breaking and suggest that the paradox can be explained by fast breaking of atomic wires and slow breaking of point contacts giving very similar breaking forces.

[1] Department of Chemistry and Biochemistry, University of Bern, Freiestrasse 3, CH-3012 Bern, Switzerland. [2] Nano-Science Center and Department of Chemistry, Universitetsparken 5, 2100 Copenhagen Ø, Denmark. [3] Institute of Materials and Environmental Chemistry, Research Centre for Natural Sciences, Hungarian Academy of Sciences, Magyar Tudósok Körútja 2, H-1117 Budapest, Hungary. [4] Department of Physics, Technical University of Denmark, 2800 Kongens Lyngby, Denmark. * These authors contributed equally to this work. Correspondence and requests for materials should be addressed to I.V.P. (email: ilya.pobelov@dcb.unibe.ch) or to G.C.S. (email: gsolomon@nano.ku.dk).

From the macroscale to the nanoscale, it has been shown that the breaking behaviour of a system depends on the rate at which force is applied, although it is not clear that the same mechanisms are in play on these vastly different length scales. At the macroscale, it is known that the yield stress of metals may depend on the strain rate[1]. At the nanoscale, it is known that the observed breaking force depends on the timescale of the breaking process[2,3]. If an applied force is loaded slowly, the structure may break spontaneously due to thermal fluctuations before any significant force is applied to the system; however if the force is loaded rapidly it is more likely that we measure the maximum force required to break the physical/chemical bond. Paradoxically, prior experiments breaking gold nanowires did not observe any clear differences in the breaking force, with all measurements reporting a breaking force of 1.2–1.8 nN (refs 4–10), despite being conducted under very different conditions.

The force-loading rate ($r_F = dF/dt$) dependence of the breaking force is well-known for biological and chemical systems, a selection of which is illustrated in Fig. 1. There are three characteristic regimes that we will refer to as spontaneous, force-assisted and activationless. In the spontaneous regime, the breaking force appears to be constant. Transitioning to the force-assisted regime, we see a sharp increase in the breaking force dependence on force-loading rate. Here, the force-loading rate is sufficiently fast for there to be an appreciable probability that the system is stretched to some extent before breaking. Finally, in the activationless regime the system breaks due to the applied force. That is, the energy barrier to breaking ($E_a$) is reduced to zero for the maximally stretched system. The activationless regime corresponds to our intuition from the macroscopic world: the breaking force is constant as the force-loading rate is always fast enough to ensure that environmental fluctuations play no role in the breaking process.

In this study, through a combination of experiment and theory, we show that the force required to break gold nanowires is indeed dependent on the force-loading rate.

We experimentally observed a near-constant breaking force over a range of force-loading rates, followed by a increase in the breaking force at higher force-loading rates. This is interpreted as breaking in the spontaneous regime with some indication of transitioning into force-assisted breaking. From simulations, we see both spontaneous and force-assisted breaking, but what appears to be the activationless regime arises from anharmonic effects. We show that different breaking structures exhibit different breaking forces while their conductance is the same, suggesting a solution to the paradox: pulling near the activationless regime at low temperatures will preferentially break atomic wires while pulling in the spontaneous regime at high temperatures will more likely break point contacts.

## Results

**The rate theory model.** A rigorous description of these three regimes arose in the aftermath of the seminal paper by Bell[3] and was presented by Evans and Ritchie[2]. Full details of the rate theory description we employ, and its relationship with the Bell–Evans model, can be found in Supplementary Note 1. We assume that the energy barrier for breaking at the initial length ($l_0$) is large compared with the thermal energy, $E_a(l_0) \gg k_B T$, and decreases linearly with the change of the system length, $l$, during the stretching

$$E_a(l) = E_a(l_0) - \alpha[l - l_0], \tag{1}$$

where $\alpha$ is a temperature-independent constant of proportionality that can be determined from the energy barrier (for further discussion see Supplementary Note 1.1). Then we can derive an expression for the critical stretching rate ($v_c$) that marks the transition from the spontaneous to force-assisted regimes:

$$v_c = k_B T R_0 / \alpha = k_B T \omega_a e^{-E_a(l_0)/k_B T} / \alpha \tag{2}$$

and the stretching rate $v_{al}$ marking transition from the force-assisted to activationless regimes:

$$v_{al} = k_B T \omega_a / \alpha \tag{3}$$

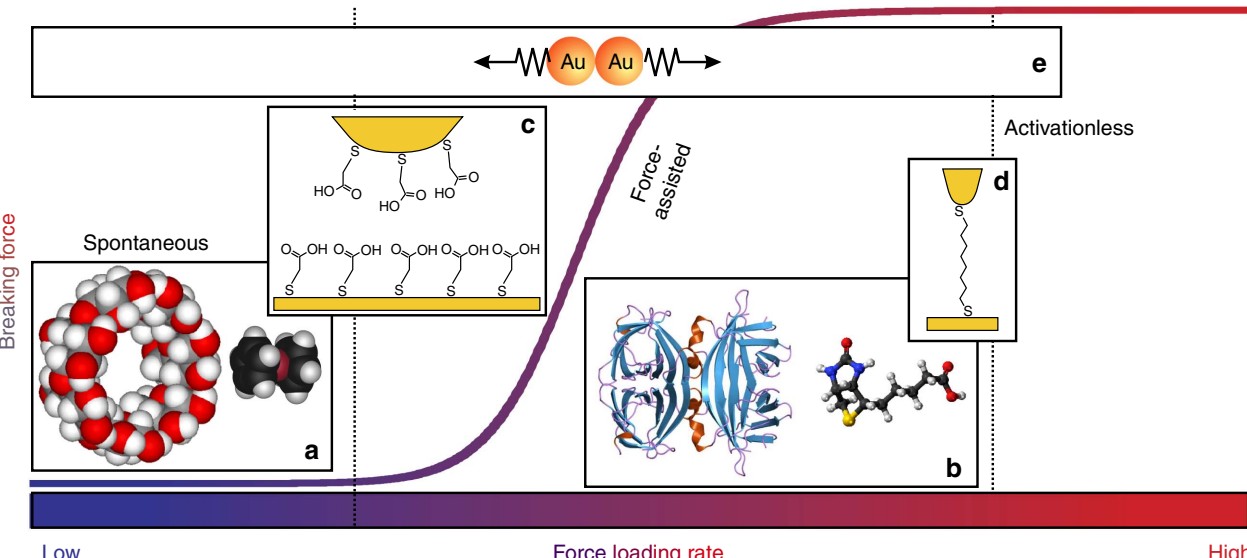

**Figure 1 | The dependence of the bond breaking force on the force-loading rate $r_F$ in three regimes (spontaneous, force-assisted and activationless)[2,26].**
(**a**) A host–guest complex formed by β-cyclodextrin (left) and ferrocene (right) breaks in a spontaneous regime with a loading rate independent breaking force[27,28]. (**b**) A plethora of complexes formed by biomolecular partners, as represented by avidin (left) and biotin (right), break in the force-assisted regime with the breaking force being a function of the force-loading rate[29]. A transition from spontaneous to force-assisted (**c**) and from force-assisted to activationless regime (**d**) was shown for the hydrogen bonding between carboxylic groups[30] and for gold–octanedithiol–gold junctions[31], respectively. (**e**) The breaking of a chemical bond formed between two gold atoms at room temperature was demonstrated to span from spontaneous to the force-assisted regime in this work. Prior cryogenic experiments[4,5] are expected to have accessed the activationless regime.

$R_0$ is the initial rate of breaking events, $R_0 = \omega_a \exp(-E_a(l_0)/k_B T)$, and $\omega_a$ is an 'attempt' frequency. The stretching rate $v$ can be converted to a force-loading rate with the knowledge of the spring constant of the cantilever in the case of an atomic force microscope (AFM) pulling experiment.

This rate theory result provides useful guidance as to which regime is active for a given set of experimental conditions. For $v \ll v_c$, the mean breaking force increases slightly with increasing stretching rate proportionally to $v$. On the other hand, for $v_c \ll v \ll v_{al}$ the breaking force increases sharply, proportional to $\ln(v)$ (Supplementary Note 1.5), and saturates at $v > v_{al}$. Furthermore, $v_c$ is highly sensitive to the height of the energy barrier (to breaking of the initial structure) and temperature, while $v_{al}$ is weakly temperature-dependent (Supplementary Note 1.6).

Single-atom thick gold nanocontacts can be readily identified in pulling experiments from either their characteristic conductance[4,6–8,11] or by direct imaging in high-resolution transmission electron microscope experiments[9,10] and they offer the possibility of tuning the system over a wide temperature range. These attributes make them likely candidates for exploring the force-loading rate dependence of breaking force, but despite prior experiments ranging from 4.2 K to room temperature, no clear upper and lower limits for the breaking force have emerged. Measurements at cryogenic temperatures (which we would not expect to be in the spontaneous regime) have reported breaking forces between 1.5 nN (ref. 4) and 1.8 nN (ref. 5), while room temperature measurements (which we expect to be in the spontaneous regime) have reported breaking forces ranging from 1.2 to 1.5 nN (refs 6–8,10).

**Identifying breaking regimes by breaking force.** When gold nanocontacts are broken, their conductance just prior to breaking can be used as a means for separating different types of breaking structures. A linear relationship was observed between conductance and breaking force, as shown in Fig. 2a. Selecting out the structures that break with $1G_0$ conductance, the dependence of the breaking force on force-loading rate could be probed and the transition from spontaneous breaking to force-assisted breaking was observed as shown in Fig. 2b. In the spontaneous regime, a breaking force of around 1.5 nN was observed, which is in good agreement with prior experimental results at 300 K and similar force-loading rates[6–8], however, much larger breaking forces were also observed at higher force-loading rates, indicating that we move into the force-assisted regime.

We note that there are challenges associated with unambiguous determination of the breaking force of a single Au–Au bond at high force-loading rates. In particular, as detailed in Supplementary Note 2.7, care has to be taken in case of the highest force-loading rates. We applied additional tests to ensure that we probe structures where the junction has relaxed sufficiently to allow us to identify the force required to break the single Au–Au bond, rather than the breaking force of larger junctions.

Recent theoretical work[12] has highlighted the role that the cantilever stiffness can play in controlling the extent to which a bond can be stretched. The highest force-loading rate was reached with a cantilever spring constant of $k_{\text{tip}} \approx 10 \, \text{N m}^{-1}$, we can compare that with the range of estimated Au–Au bond spring constant of $k_{\text{Au}} \approx 2\text{–}10 \, \text{N m}^{-1}$ (ref. 13). This puts the system in the regime where both the cantilever and bond are stretched, possibly limiting the extent to which system can move into the force-assisted regime.

The transition from spontaneous to force-assisted breaking can be reproduced in a model theoretical system, shown in Fig 3a. These results comprise both data points from a large number of repeated pulls in molecular dynamics (MD) simulations of the

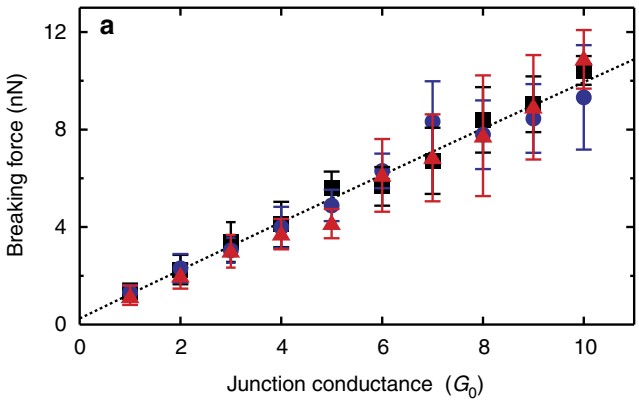

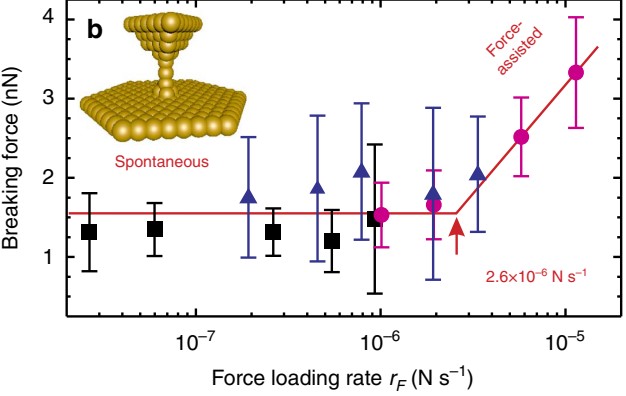

**Figure 2 | The breaking force of gold nanocontacts measured by current-sensing AFM at room temperature.** The symbols are the mean value, error bars are the standard deviation (s.d.). (**a**) The conductance dependence of the breaking force obtained by employing cantilevers with spring constant $k \approx 5 \, \text{N m}^{-1}$. The stretching rate $v$ was 10 (■), 50 (●) or 100 (▲) $\text{nm s}^{-1}$. The line represents the linear fit through all data points. (**b**) Results obtained for nanocontacts with the conductance $1G_0$ employing cantilevers with spring constants $k \approx 5 \, \text{N m}^{-1}$ (■), $k \approx 10 \, \text{N m}^{-1}$ (●) and $k \approx 40 \, \text{N m}^{-1}$ (▲). The lines are the average force in the spontaneous breaking regime ($r_F < 2 \times 10^{-6} \, \text{N s}^{-1}$) and a linear fit of force versus log $r_F$ dependence in the force-assisted breaking regime ($r_F > 5 \times 10^{-6} \, \text{N s}^{-1}$). Extrapolating these two lines gives $r_F \approx 2.6 \times 10^{-6} \, \text{N s}^{-1}$ for the transition between the regimes.

six-atom wire shown in the inset and also a fitting curve from rate theory. While in all cases the transition from the spontaneous to force-assisted regimes is clearly visible, the magnitude of the breaking force differs significantly between theory and experiment. At this point, we note that the horizontal axes on Figs 2b and 3a,b show different variables. For the experiments the natural variable is the force-loading rate, the product of the cantilever stiffness and the probe retraction rate. For the simulations the natural variable is the stretching rate.

As the magnitude of the breaking force in the spontaneous regime is strongly dependent on the initial structure, this raises the question of whether the $1G_0$ structures observed in experiment are accurately represented by a single-atom thick wire. The simulations agree well with prior simulations at the same level of theory at high stretching rates[4], but we are hesitant to compare the lower breaking forces in the spontaneous regime due to the strong dependence on initial structure.

**Stretching distance and survival probability.** Before we probe further into the discrepancies in the breaking force s between prior

experiments and the results reported here at high force-loading rate, we can also examine other variables related to the two regime transitions that can be obtained from simulation: the stretching distance and survival probability ($P(t)$, the fraction of all simulations at a given stretching rate that did not break after a time, $t$), as shown in Fig. 3b,c. Clear signatures of the transition from spontaneous to force-assisted breaking are also evident in these variables. The stretching distance dependence on stretching rate has been previously observed experimentally[14] and three regimes were observed in that case although no forces were reported. In our calculations, the activationless regime is only reached at stretching rates beyond the rates shown in Fig. 3a. However, a breaking force of about the same value as the activationless breaking force is already reached at the stretching rate $0.1\,\mathrm{m\,s^{-1}}$ due to anharmonic effects (see Supplementary Note 3.5 for further discussion on this point). The survival probability below $10^{-5}\,\mathrm{m\,s^{-1}}$ is independent of stretching rate, indicating that temperature alone is controlling the kinetics of the breaking process at these rates. At higher rates, we see a dependence that indicates the applied force is resulting in a reduction of the energy barrier to breaking and as a consequence the kinetics of the breaking process change (Supplementary Note 1.2).

**Effects of temperature.** Turning to the question of the discrepancies in the forces, rate theory provides guidance as to the system parameters that control both the magnitude of the

observed breaking force and the transitions between regimes. Temperature appears explicitly in equation (2), and while it does not change the breaking force in the high- and low-stretching rate limits to a great extent, it does shift $v_c$ significantly. The transition to the activationless regime, however, is much less sensitive to temperature. A full discussion of the temperature dependence of the model parameters and transitions between regimes is given in Supplementary Note 1.6. In our simulations, for relatively small changes in temperature, we can see that $v_c$ drops by approximately one order of magnitude with each 30 K decrease in temperature as shown in Fig. 4a. With the reduction to 4 K, we can estimate that $v_c$ decreases to the order of $10^{-500}\,\mathrm{m\,s^{-1}}$, as shown in Supplementary Note 3.6.

Such a dramatic reduction in $v_c$ means that clearly only the activationless and, possibly, force-assisted regimes are accessible at cryogenic temperatures. The question then is: if only a small fraction of the force-assisted regime is accessible, to what extent are forces below the activationless limit observable at cryogenic temperatures? We can answer this question by estimating the most probable breaking length at 4 K over the full range of pulling speeds, as shown in Supplementary Note 3.6. From this

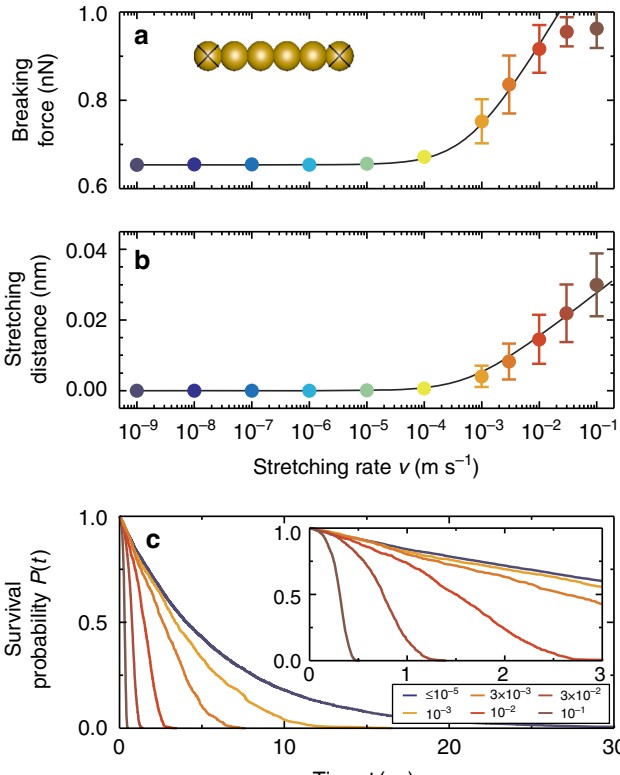

**Figure 3 | Simulation results for a six-atom gold nanowire at varying stretching rates.** Mean value (symbols) and the s.d. (error bars) of the breaking force (**a**) and stretching distance (**b**) as a function of stretching rate from MD simulations (at 300 K) of the structure in the inset. The crossed atoms are constrained during the MD steps of the simulation. (**c**) Survival probabilities $P(t)$ at various stretching rates ($\mathrm{m\,s^{-1}}$) from MD simulations. All stretching rates below $10^{-5}\,\mathrm{m\,s^{-1}}$ show the same survival probability (blue line).

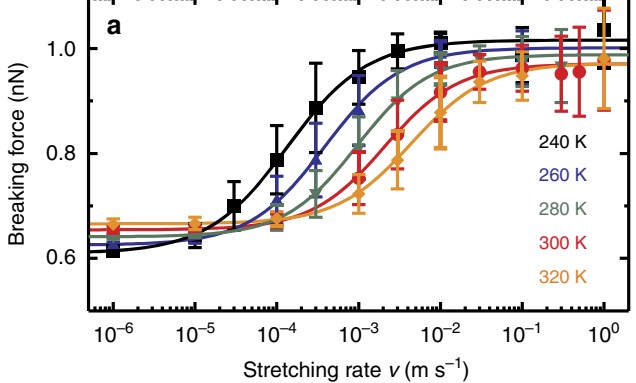

| b | Max. force (nN) | | $G(G_0)$ |
|---|---|---|---|
| Structure | EMT | DFT | |
| | 1.1 | 1.5 | 0.86 |
| | 2.2 | 2.9 | 0.81 |
| | 2.2 | 2.8 | 1.09 |
| | 2.0 | 3.2 | 1.18 |

**Figure 4 | Simulated variation in breaking force as a consequence of varying temperature or structure.** Temperature and structure (and thereby $E_a$) are important parameters for controlling both the magnitude of the breaking force and $v_c$. (**a**) The breaking of the six-atom wire from MD simulations (symbol is the mean value, error bar is the s.d.), with fit lines to guide the eye, for a range of temperatures. (**b**) The maximum breaking force for a range of point contacts calculated using EMT and density functional theory (DFT) in the activationless regime. The junctions are stretched by displacing the crossed atoms. The calculated zero-bias conductance is also shown.

calculation it is clear that while the most probable breaking length deviates from the activationless limit, it only does so by tenths of an Å. This means that irrespective of the pulling speed, at 4 K the breaking force is essentially indistinguishable from the activationless limit.

More significant in controlling the magnitude of the breaking force is $E_a$, which necessarily depends on the structure that is being broken. We can also probe the influence of structure in simulation; however it is unfortunately not possible to map the entire range of stretching rates. The small six-atom system and our theoretical method were chosen to allow us to access a full eight orders of magnitude in stretching rates, with a statistically significant number of pulls in each case. This is not possible with larger gold structures. We can, however, probe the maximum breaking force for the activationless regime for a range of different structures. Figure 4b shows the maximum breaking forces calculated for a range of point-contact structures using both effective medium theory (EMT) and density functional theory, further details are given in Supplementary Notes 3.7 and 3.8. While the linear chain exhibits a maximal breaking force that agrees well with measurements at cryogenic temperatures, the point-contact structures have significantly higher breaking forces. This variation in breaking force for $1G_0$ structures has also previously been reported from MD simulations[15].

## Discussion

Returning to the apparent paradox from the literature, that the breaking force observed for the spontaneous regime at room temperature seemed to be the same as the breaking force observed for the activationless regime at cryogenic temperatures, we can now see a resolution. The paradox exists if we assume that it is the same types of structures breaking in all cases, but it is certainly likely that a $1G_0$ structure at room temperature is, on average, more point-contact-like (that is, it is less likely that a multi-atom linear chain is formed prior to breaking), while a $1G_0$ structure at cryogenic temperatures is more linear-chain-like. The confusion arose because of the coincidence that point-contact structures in the spontaneous regime appear to break with very similar breaking force to linear chains in the activationless regime.

High-resolution transmission electron microscope experiments provide some more direct evidence for the hypothesis that structures observed in cryogenic and room temperature measurements are not the same. While cryogenic experiments clearly show single-atom thick chains, these were only seen in the presence of impurities at room temperature[9] and otherwise a range of point-contact-like structures were observed[10].

While we are reasonably confident in excluding the possibility that a significant fraction of linear chains are formed at room temperature, this assumption will not hold as the temperature is decreased significantly. In that case, one can imagine that changing the pulling speed could also change the distribution of breaking structures, for example between the four structures illustrated in Fig. 4b, and lead to the average breaking force either increasing or decreasing with force-loading rate.

These results also suggest caution when comparing the breaking force measured under different conditions. The temptation is always to attribute differences in breaking force to system differences (or experimental details if the systems should be the same); however, these results give us the strong signal that we must also be sure that the measurements are in the same breaking regime before such a comparison is made. For our model theoretical system, rate theory suggests that $v_c$ will drop by approximately an order of magnitude with a 0.05 eV increase in the initial barrier height (for example, by changing the nature of the system) or a 30 K temperature decrease. This raises the very

real question of whether the results shown in Fig. 2a are all in the same breaking regime. The clear linear trend would tend to suggest they are, and we are able to obtain rough force-loading rate dependence for the $2G_0$ data (as shown in Supplementary Note 2.7) that confirms that this system is also in the spontaneous regime. It is impossible to rule out, however, that the spread in the data at higher conductance values might, in part, arise from a range of structures that transition into the force-assisted regime.

Together these results suggest that there is more to be observed in gold nanowires—with lower and higher breaking forces than the 1.2–1.5 nN that had previously been accepted. It seems unlikely that forces below the activationless limit can be accessed at 4.2 K, but between these cryogenic temperatures and room temperature some significant variation is possible. The intuition that we can probe the nature of a gold–gold bond by breaking it needs to be refined with the knowledge that not all $1G_0$ structures are created equal and thermal fluctuations may play a significant role in the breaking process.

## Methods

**Experimental.** Experimental breaking force of Au–Au nanocontacts was determined by current-sensing force spectroscopy[11,16,17] as described in Supplementary Note 2. The force $F$ acting on the cantilever and the probe-substrate conductance $G$ were recorded during the repeated approach/withdrawing of the gold-covered AFM probe to/from the gold substrate in decane. Au nanocontacts created upon the probe pulling demonstrate well-defined quantized conductance of $G \approx NG_0$, where $G_0 = 77.5\,\mu S$ is the quantum of conductance and $N$ is the number of gold atoms in the narrowest part of the nanocontact[18]. The force-conductance traces that displayed conductance plateaus at $G \approx NG_0$ just before the contact break were analysed to obtain the experimental mean breaking force for a gold–gold contact with a given $N$. In particular to determine the breaking force of a single Au–Au bond, we only included traces with $G \approx G_0$ before the break.

**Simulations.** Langevin MD simulations were performed in the atomistic simulation environment[19,20]. The energy and force were calculated using EMT[21,22]. The EMT potential includes many-atom interactions and have previously been shown to describe the formation of atomic chains in gold nanocontacts[4]. We used a Au chain with six atoms as a model system. The initial structure was a completely linear chain with the two end atoms fixed at a distance of 1.22 nm. Prior to the simulation the structure was relaxed. The pulling was applied by running MD for a given number of steps, then moving one of the end atoms, fixing the end atom again and repeating. This model necessarily assumes that the rest of the tip and surface is effectively rigid and does not relax either during the pulling or on the timescale of spontaneous breaking. It remains a question for future work as to when this assumption might break down. The temperature was varied from 240 to 320 K while the stretching rate varied from $10^{-9}$–$10^{-1}\,m\,s^{-1}$ (see Supplementary Note 3.1 for details). The MD time step was 0.5 fs and the stretching interval was $10^{-4}$ Å. One thousand (1000) traces were generated for each set of conditions. The force acting on nanowire was calculated as $F = (F_1 - F_6)/2$, where $F_1$ and $F_6$ are the forces on the terminal atoms of the wire. The breaking force was found from a linear fit of the simulated forces (more details are given in Supplementary Note 3.2). The chain was considered broken when any two neighbouring atoms were further than 0.4 nm apart.

The conductance values for the point contacts were calculated with desity functional theory in Atomistix ToolKit version 2014.3 (refs 23,24). Additional details for the point-contact calculations are available in Supplementary Notes 3.7 and 3.8.

To calculate energy barrier for rate theory, we applied the nudged elastic band method[25] while the attempt frequency was used as a fitting parameter (Supplementary Note 3.3).

**Data availability.** The data that support the findings of this study are available from the corresponding authors on request.

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

## Acknowledgements

This work was supported by The European Union Seventh Framework Programme (FP7/ 2007–2013) under the ERC grant agreement no. 258806, the Danish Council for Independent Research—Natural Sciences, the Carlsberg Foundation, the Swiss National Science Foundation (200020_144471, NFP 62, Sinergia CRSII2 126969/1), Swiss Commission for Technology and Innovation (13696.1), COST Action TD 1002, OTKA (105735), and the bilateral mobility programme of HAS (SNK-61/2013).

## Author contributions

I.V.P., G.C.S. and T.W. conceived the original concept. I.V.P., G.M. and T.W. designed the experiment; G.M. built employed instrumentation; I.V.P. and K.Y. carried out CSAFM experiments; I.V.P. analysed experimental data. K.P.L., M.S. and G.C.S. developed the model for MD simulations; K.P.L. performed the MD simulations; K.P.L. and A.J. analysed simulated data. I.V.P., M.S. and K.W.J. elaborated the rate theory model. I.V.P., K.P.L. and G.C.S. prepared the manuscript using feedback from other authors.

## Additional information

**Competing interests:** The authors declare no competing financial interests.

**Reprints and permission** information is available online at http://npg.nature.com/ reprintsandpermissions/

**Publisher's note**: 

