## [Peer Review File · Nature Communications]

Reviewers' comments:

Reviewer #1 (Remarks to the Author):

The manuscript by Pobelov et al explores the dependence of the breaking force of a single gold-gold bond on the breaking rate. They combine experiments performed in a current-sensing AFM with theoretical calculations using effective medium theory (EMT) and density functional theory (DFT). The main claim of the paper is that the breaking force can be much higher (more than a factor of 2 higher) for faster breaking rates than for slower breaking rates. This is at variance with previously reported experimental results from different groups at cryogenic temperatures and at room temperature. The main claim of the paper is that they observe experimentally and explain theoretically the transition from the spontaneous breaking regime to the activationless breaking regime. This is an interesting and novel claim, however I still have serious doubts about the quality of the experimental data and find the theoretical explanation unconvincing. Consequently I find the paper unsuitable for publication in Nature Communications.

From the experimental point of view, the main claim of the paper relies on just two data points with large error bars (after the points for the largest force-loading rates, present in the previous version of the manuscript have been suppressed following a more careful selection process for the valid data traces). Furthermore, the validity of this two points is still questionable since contamination and/or the dynamics of the force sensor in the liquid environment could be affecting the results, as I will discuss below.

1. The authors have discarded unclear traces (as detailed in SI section B4), but this procedure which is based in the absence of conductance plateaus below 1 G0 does not really guarantee the absence of contamination that could be contributing to breaking forces for the one-atom contact. For a soft cantilever the accumulation of strain before the breaking can lead to a rapid snap-off which will prevent the observation of any contamination related plateau explaining why "the data sets measured by cantilevers with $k \sim 40$ N/m displayed much higher t_n than for the other cantilevers". Since the harder cantilevers would not be more contaminated than the softer ones, this indicates that the contamination is more likely to be detected with the softer cantilevers, and this effect will be more noticeable for the higher loading rates. Consequently, contamination cannot be ruled out for the higher loading rate experiments.

2. Looking carefully at figure S12, one can observe that (a) the conductance decreases sharply at time=0, however the force trace starts increasing at time=0.1 ms (in particular the blue and black traces that show a change of slope, the others are even less reliable as they seem part of an oscillation). What is causing this delay? The authors state that "a careful inspection of yielding events observed on conductance and force traces does not show signs of desynchronization between [the] two signals", but figure S12 shows otherwise. Is this delay due to the AFM electronics or is the liquid environment affecting the measurements? Note that the effect of the liquid on the dynamics of tip-substrate separation might be very important since the motion will be in the very low Reynolds number regime, which implies that viscous effects will dominate ($U \sim 200$ nm/s, $L \sim 1$ nm, kinematic viscosity of decane $\sim 10^{-6}$ m²/s, yield $Re \sim 10^{-9}$).

From the theoretical point of view, I must remark that the comparison between the experimental force loading rates and the theoretical stretching rates is very difficult (this is partially commented in page 4). The simulations assume that the atoms close to the contact separate in a controlled manner (as indicated in Figure 4) but this is completely unrealistic since in the breaking of a metallic contact the elasticity of the whole junction is playing a role (that is even in an STM experiment the force loading rate will be the relevant magnitude (not the stretching rate). Couldn't the calculations be improved by adding a spring in series with the junction to account for this elasticity. In page 5 it is stated that "at 4K we expect the entirety of the experimentally accessible range of stretching rates to fall in the activationless regime", why there are no simulations for this temperature?

Reviewer #2 (Remarks to the Author):

I have carefully read the 3rd revision to this manuscript, as well as the authors' response letter. The new revised version addressed all of my concerns. I also feel that authors' responses to the 1st and 3rd referees are serious and candid. Overall, this is a very nice piece of experimental-computational work that successfully addressed a fundamental question in this field, which is in much need to explore. I do not have any reservation regarding the publication of this work in the Nature Communications.

Reviewer #3 (Remarks to the Author):

The paper addresses the problem of the breaking force in Au atomic contacts. The authors point at a paradox posed by the observation of breaking forces that appear to be independent of the speed of breaking, and independent of temperature. On the other hand, simulations suggest that the observed force should depend on the loading rate within the range of experimental parameters. The solution the authors propose is that there may be a coincidence: at low temperatures the atomic configurations probed differ from those at room temperature.

Although the authors cannot determine the actual configurations in the experiment, the argumentation is sufficiently plausible.

The paper is well written and can be accepted for publication.

We would like to thank all referees for their review of our manuscript. In particular we are grateful to the thoughtful comments of reviewer 1, we agree that our manuscript is improved by these revisions.

Reviewer #1 (Remarks to the Author):

The manuscript by Pobelov et al explores the dependence of the breaking force of a single gold-gold bond on the breaking rate. They combine experiments performed in a current-sensing AFM with theoretical calculations using effective medium theory (EMT) and density functional theory (DFT). The main claim of the paper is that the breaking force can be much higher (more than a factor of 2 higher) for faster breaking rates than for slower breaking rates. This is at variance with previously reported experimental results from different groups at cryogenic temperatures and at room temperature. The main claim of the paper is that they observe experimentally and explain theoretically the transition from the spontaneous breaking regime to the activationless breaking regime. This is an interesting and novel claim, however I still have serious doubts about the quality of the experimental data and find the theoretical explanation unconvincing. Consequently I find the paper unsuitable for publication in Nature Communications.

From the experimental point of view, the main claim of the paper relies on just two data points with large error bars (after the points for the largest force-loading rates, present in the previous version of the manuscript have been suppressed following a more careful selection process for the valid data traces). Furthermore, the validity of this two points is still questionable since contamination and/or the dynamics of the force sensor in the liquid environment could be affecting the results, as I will discuss below.

1. The authors have discarded unclean traces (as detailed in SI section B4), but this procedure which is based in the absence of conductance plateaus below 1 G0 does not really guarantee the absence of contamination that could be contributing to breaking forces for the one-atom contact.

While “guaranteeing” the absence of contamination might not be possible (arguably for any experiment under ambient conditions), we certainly take the utmost care to analyze our data correctly and clearly describe the analysis procedure in detail in supporting information. All experiments were carried out in the same environment, and it is clear that the results obtained for force loading rates below $2 \cdot 10^{-6}$ N/s correspond well to results from other research groups obtained under similar conditions. The trend shown in Figure 2b is also clear. The error bars, estimated as described in SI, are rather large, but the difference between mean values at the right side of Figure 2b is higher than error bars. We do not see how contaminations can affect results of only exactly 2 experiments carried out with the fastest force loading rates from more than 20 experiments in such way that the breaking force for two points at highest force loading rate produces a significant increasing trend.

For a soft cantilever the accumulation of strain before the breaking can lead to a rapid snap-off which will prevent the observation of any contamination related plateau explaining why “the data sets measured by cantilevers with $k \sim 40$ N/m displayed much higher t_n than for the other cantilevers”.

The accumulation of the strain was indeed an issue with soft ($k \sim 5$ N/m) cantilevers, but it is not at all relevant for hard cantilevers. Quoting supporting information, page 13, “prior to the further analysis we removed from the experimental data sets traces showing the decrease of the conductance from $G > 10.5 G_0$ to $G < 0.1 G_0$ in less than 0.1 ms.”. The number of traces rejected according to this criterion was higher for softer cantilever. However, to obtain our final results we analyzed traces that showed gold conductance plateaus and, most importantly, $1G_0$ conductance plateaus, and did not have such “rapid snap-off” features. Therefore we believe the reviewer’s argumentation is not applicable in this case.

As we also stated in supporting information, page 15, we attribute higher t_n “to the contribution of tunnelling current measured after the breaking of the gold-gold contact”. To clarify this point in detail, we added in section B.4 2D conductance-time and conductance-distance histograms calculated from traces measured with hard and soft cantilever with stretching rates of 10 and 100 nm/s (Figures S7 and S9). In their discussion, we explained in details that the conductance-distance dependence observed for hard cantilevers after the contact break is very similar to the one obtained in break-junction type experiments and well known to the authors.

We also note that in the previous publications of Bern group (Refs. 17 and 21) we successfully measured conductance and breaking force of gold-molecule-gold junctions with soft cantilevers. This indicates that we would measure the conductance plateaus of contaminants as well, if they are present in the junction.

Since the harder cantilevers would not be more contaminated than the softer ones, this indicates that the contamination is more likely to be detected with the softer cantilevers, and this effect will be more noticeable for the higher loading rates. Consequently, contamination cannot be ruled out for the higher loading rate experiments.

We find the statement of reviewer that “the contamination is more likely to be detected with the softer cantilevers” contradicting to the above statement that “For a soft cantilever the accumulation of strain before the breaking can lead to a rapid snap-off which will prevent the observation of any contamination related plateau”. As we stated above, we do not understand how contaminations can affect only two results obtained at higher force loading rates in a systematic way.

2. Looking carefully at figure S12, one can observe that (a) the conductance decreases sharply at time=0, however the force trace starts increasing at time=0.1 ms (in particular the blue and black traces that show a change of slope, the others are even less reliable as they seem part of an oscillation). What is causing this delay? The authors state that “a careful inspection of yielding events observed on conductance and force traces does not show signs of desynchronization between [the] two signals”, but figure S12 shows otherwise. Is this delay due to the AFM electronics or is the liquid environment affecting the measurements?

Note that the effect of the liquid on the dynamics of tip-substrate separation might be very important since the motion will be in the very low Reynolds number regime, which implies that viscous effects will dominate ($U \sim 200 \text{ nm/s}$, $L \sim 1 \text{ nm}$, kinematic viscosity of decane $\sim 10^{-6} \text{ m}^2/\text{s}$, yield $Re \sim 10^{-9}$).

We carefully reinspected our experimental curves and summarized results in the last paragraph of section B.7 in SI. First, the difference between the last point before the sharp conductance decrease and the inflection point on corresponding force curve is not systematic and can be both negative and positive. For example, following figure shows a zoom of 3 other curves from new Figure S14. They all shows start of force increase at $t=0$. Second, the apparent difference between the “last point of conductance plateau” and “point where force starts to increase” can be caused by the subjectivity of their determination in real experimental curves. However, we do not exclude an instrumental factor in such delay, as described in SI. In either case, the difference does not exceed 0.15 ms, which is significantly smaller than the rise time of the force signal and thus does not affect our conclusions.

On the other hand, we think that the “rise time” of force signal that is equal to 0.5 ms and observed independently on the experimental conditions is related to the relaxation of cantilever in the liquid environment, as pointed out by the reviewer. We specified this fact by modifying the sentence

“A possible explanation for the finite rise time of the force immediately after the contact break is an intrinsic, stretching rate independent relaxation of the cantilever” as

“A possible explanation for the finite rise time of the force immediately after the contact break is an intrinsic, stretching rate independent relaxation of the cantilever in the liquid environment”.

From the theoretical point of view, I must remark that the comparison between the experimental force loading rates and the theoretical stretching rates is very difficult (this is partially commented in page 4). The simulations assume that the atoms close to the contact separate in a controlled manner (as indicated in Figure 4) but this is completely unrealistic since in the breaking of a metallic contact the elasticity of the whole junction is playing a role

(that is even in an STM experiment the force loading rate will be the relevant magnitude (not the stretching rate). Couldn't the calculations be improved by adding a spring in series with the junction to account for this elasticity.

We agree with the referee that the elasticity of the whole junction plays a role in how a junction breaks. But in the case we are interested in, where there is just a single atom in the smallest cross-section, we do not expect any spring added in series to have a qualitative effect on the results of the simulation. If we looked at force versus system length, adding a spring would effectively be a rescaling of the length, but with the same forces recorded.

See Sørensen, M. R., Brandbyge, M., & Jacobsen, K. W. (1998). Mechanical deformation of atomic-scale metallic contacts: Structure and mechanisms. *Physical Review B*, 57(6), 3283.

<https://doi.org/10.1103/PhysRevB.57.3283>

In figure 13 in the above reference it is shown that the effect of adding a spring is significant when a rearrangement in the junction occurs, but appears as a simple scaling factor during elastic loading.

The simulations aim to show that we indeed can see two regimes of breaking force when spanning orders of magnitude in stretching rates. We are not trying to make any prediction on what force loading rate an experimental setup should aim for to see this transition. This we do purely from the experimental data.

In page 5 it is stated that “at 4K we expect the entirety of the experimentally accessible range of stretching rates to fall in the activationless regime”, why there are no simulations for this temperature?

This question has prompted a lot of thought and discussion during the course of our revisions, and we have now added a significant amount of new material being integrated throughout the S.I. as well new text in the main manuscript.

While we could certainly perform pulling simulations for the fastest pulling speeds at 4K, it is not possible for us to reach the slowest speeds (ie the full range) due to the extremely long simulation time required. We can address this issue from a theoretical point of view with the help of rate theory.

In short, at these low temperatures the ratio of energy barrier to thermal energy is so high that the breaking can only occur when stretching significantly reduces the barrier. In the new sections we show that at 4K, stretching at nanometers per second, we would only expect the junction to break when it is just 2 pm from the point where the barrier goes to zero. That is, while it is possible to technically be in the force-assisted regime at 4K for very slow pulling speeds, the breaking force will be indistinguishable from the activationless limit. With this result in mind, we agree that the statement highlighted by the referee was not strictly correct, and have thus clarified this result.

The changes are:

- In the supplementary material:
 - Additional material in new sections A.4, A.5 and A.6 detailing the rate theory results for: when the system will transition to the activationless limit; the mean stretching time, stretching length and breaking force across all three regimes; and the temperature dependence of the model parameters.
 - New section C.6 detailing the most probably stretching distance at cryogenic temperature from simulations
- In the main text, we have removed the offending sentence and have added additional paragraphs on page 5 to clarify the temperature dependence of the breaking regimes.

Reviewer #2 (Remarks to the Author):

I have carefully read the 3rd revision to this manuscript, as well as the authors' response letter. The new revised version addressed all of my concerns. I also feel that authors' responses to the 1st and 3rd referees are serious and candid. Overall, this is a very nice piece of experimental-computational work that successfully addressed a fundamental question in this field, which is in much need to explore. I do not have any reservation regarding the publication of this work in the Nature Communications.

We thank the reviewer for their comments.

Reviewer #3 (Remarks to the Author):

The paper addresses the problem of the breaking force in Au atomic contacts. The authors point at a paradox posed by the observation of breaking forces that appear to be independent of the speed of breaking, and independent of temperature. On the other hand, simulations suggest that the observed force should depend on the loading rate within the range of experimental parameters.

The solution the authors propose is that there may be a coincidence: at low temperatures the atomic configurations probed differ from those at room temperature.

Although the authors cannot determine the actual configurations in the experiment, the argumentation is sufficiently plausible.

The paper is well written and can be accepted for publication.

We thank the reviewer for their comments.

REVIEWERS' COMMENTS:

Reviewer #1 (Remarks to the Author):

The authors have answered satisfactorily the issues raised by the data at the fastest stretch rates by eliminating these data, since they were clearly beyond the measuring capability of their system. The last two rightmost points indicate an increase in the breaking force.

However, the arguments in page 6 to explain the “apparent paradox”, that the force observed for the breaking at low temperatures (allegedly in the activationless regime) and that of the force measured at room temperature (allegedly in the spontaneous regime) seemed to be the same, offer a simpler explanation for the observed results. The authors show convincingly that there may be different structures with a conductance of 1 G0 but different breaking forces (Fig. 4), and they argue that narrower structures (chain-like) will be more likely at low temperatures, and more point-contact-like, at room temperature. There is evidence that this is the case in the low temperature experiments that do observed the formation of atomic chains. However, it is somewhat naive to assume that the structure will not be also influenced by the stretching rate. One may well argue that contacts breaking at lower rates are likely to have narrower structures because they have more time to evolve, while those breaking at higher rates will have a tendency to have more compact structure when breaking. This a plausible (and simpler) explanation to the observed results, and to the apparent paradox.

This explanation of the experimental results is incompatible with the interpretation given by the authors, which assumes that the structure of the contact at breaking is independent of the rate. The only way to demonstrate that the author’s interpretation is correct would be to reach the activationless regime.

I cannot recommend publication of this paper in Nature Communications in its present form.

Editorial Changes

We have been through the manuscript and supporting information and endeavored to make all the required changes. All changes are marked in blue.

Reviewer #1

The authors have answered satisfactorily the issues raised by the data at the fastest stretch rates by eliminating these data, since they were clearly beyond the measuring capability of their system. The last two rightmost points indicate an increase in the breaking force.

However, the arguments in page 6 to explain the “apparent paradox”, that the force observed for the breaking at low temperatures (allegedly in the activationless regime) and that of the force measured at room temperature (allegedly in the spontaneous regime) seemed to be the same, offer a simpler explanation for the observed results. The authors show convincingly that there may be different structures with a conductance of 1 G₀ but different breaking forces (Fig. 4), and they argue that narrower structures (chain-like) will be more likely at low temperatures, and more point-contact-like, at room temperature. There is evidence that this is the case in the low temperature experiments that do observed the formation of atomic chains. However, it is somewhat naïve to assume that the structure will not be also influenced by the stretching rate.

This is an interesting question posed by the referee.

In principle, two mechanisms can produce significant variation of junction: surface diffusion and electromigration (or electric field induced surface migration) of gold atoms. The latter requires application of potential difference of at least few volts between electrodes, while we applied between the probe and the sample the bias voltage of only 0.13 V. Thus, we can exclude migration due to the potential difference. We consider diffusion further below.

One may well argue that contacts breaking at lower rates are likely to have narrower structures because they have more time to evolve, while those breaking at higher rates will have a tendency to have more compact structure when breaking.

This is not necessarily correct. While the slower pulling speed could certainly allow atoms to diffuse out of the junction (leading to narrower structures), the opposite is also true – slower pulling speeds could allow atoms to diffuse in to the junction. The question is which of these processes is energetically favorable. Answering this question quantitatively is beyond the scope of our study.

What is possible, however, is to consider whether a changing distribution of structures at room temperature could lead to the experimental results observed. If we assume that the full range of structures shown in figure 4 are accessible, then one can anticipate that if the probability of breaking from a 6 atom gold wire shifts appreciably with force loading rate, the observed breaking force could either increase or decrease. The problem is that, to the best of our knowledge, there is no good evidence that linear chains are pulled out at room temperature. If we consider only a variable distribution of point contact structures, we cannot obtain a systematic variation in breaking force as the

difference in breaking force between the point contact structures is much less than the experimental error bar, which is in turn less than the difference between experimental mean values, and therefore is not significant.

We absolutely agree with the referee that we have no reason to assume the distribution of breaking structures remains constant over the range of pulling speeds without more detailed evidence of the range of energy barriers involved. But, equally, we are confident that this distribution of structures is all point contact structures at room temperature. That being the case, we do not believe this explains the results obtained.

We think this may become relevant at lower temperatures so have added a paragraph to this effect in the discussion.

Text added:

“While we are reasonably confident in excluding the possibility that a significant fraction of linear chains are formed at room temperature, this assumption will not hold as the temperature is decreased significantly. In that case, one can imagine that changing the pulling speed could also change the distribution of breaking structures, for example between the four structures illustrated in Figure 4(b), and lead to the average breaking force either increasing or decreasing with force loading rate.”

This a plausible (and simpler) explanation to the observed results, and to the apparent paradox.

This explanation of the experimental results is incompatible with the interpretation given by the authors, which assumes that the structure of the contact at breaking is independent of the rate.

The only way to demonstrate that the author’s interpretation is correct would be to reach the activationless regime.

I cannot recommend publication of this paper in Nature Communications in its present form.

Additional Changes

Since our submission of the previous version of this manuscript, we have become aware of a new publication from Mark Hybertsen that is relevant for this work. We have added a reference to this manuscript as well as a paragraph putting our results in context in the main text and some small changes in the SI. Finally, we have made very small grammatical changes after a final proof-reading by all authors.